# OpenReview forum: "LLM Targeted Underperformance Disproportionately Impacts Vulnerable Users"
_NeurIPS.cc/2024/Workshop/SafeGenAi — SafeGenAi Poster_

### Official Review · Reviewer_xGsx · 2024-10-09
**Review of LLM Targeted Underperformance Disproportionately Impacts Vulnerable Users**

**Rating:** 4
**Confidence:** 4

**Review:**

Pros:

[1] The results of this study and the design of adding short biographies as context or persona are interesting as they present a new way to assign a persona.

[2] The contribution of the modifying the education level is innovative and a new contribution, as most persona based injection works focus on adding a limited set of personas, limited to race, gender, political affiliation etc.

[3] The experimentation landscape is thorough as the authors have looked recent LLMs including Claude, which ensures there experiments are generalizable

Cons:

[1] The prompt used to generate bios is not controllable. While the authors claim that they study primarily the education level, English proficiency and the nation of origin, the prompt used to generate the examples, adds other features that are not accounted for. This is visible in the examples which contain information about the persona's interests. This additional information is not accounted for in the results, and thus reduces the author's ability to make a causal claim about LLM being biased towards "vulnerable users".

[2] The prompt used to inject the persona, as provided in the example, is not clear. By not providing the model instruction on whether the bio is of a user persona or a model persona, the model might get confused hence cause deviation and bias. Authors should refer to:
Park, Minju, et al. "Empowering personalized learning through a conversation-based tutoring system with student modeling." Extended Abstracts of the CHI Conference on Human Factors in Computing Systems. 2024. on how persona is added to the model

---

### Decision · Program_Chairs · 2024-10-09

Accept (Poster)